# Coherent interaction-free detection of microwave pulses with a superconducting circuit

Shruti Dogra [1] ✉, John J. McCord [1] ✉ & Gheorghe Sorin Paraoanu [1] ✉

The interaction-free measurement is a fundamental quantum effect whereby the presence of a photosensitive object is determined without irreversible photon absorption. Here we propose the concept of coherent interaction-free detection and demonstrate it experimentally using a three-level superconducting transmon circuit. In contrast to standard interaction-free measurement setups, where the dynamics involves a series of projection operations, our protocol employs a fully coherent evolution that results, surprisingly, in a higher probability of success. We show that it is possible to ascertain the presence of a microwave pulse resonant with the second transition of the transmon, while at the same time avoid exciting the device onto the third level. Experimentally, this is done by using a series of Ramsey microwave pulses coupled into the first transition and monitoring the ground-state population.

Since the inception of quantum mechanics, the quest to understand measurements has been a rich source of intellectual fascination. In 1932 von Neumann provided the paradigmatic projective model[1] while in recent times a lot of research has been done on alternative forms and generalizations such as partial measurements and their reversal[2–5], weak measurements[6–9] and their complex weak values[10,11], observation of quantum trajectories[12,13], and simultaneous measurements of non-commuting observables[14–16].

The interaction-free measurements belong to the class of quantum hypothesis testing, where the existence of an event (for example the presence of a target in a region of space) is assessed. In a nutshell, the interaction-free detection protocol[17] provides a striking illustration of the concept of negative-results measurements of Renninger[18] and Dicke[19]. The very presence of an ultrasensitive object in one of the arms of a Mach-Zehnder interferometer modifies the output probabilities even when no photon has been absorbed by the object. The detection efficiency can be enhanced by using the quantum Zeno effect[20] through repeated no-absorption "interrogations" of the object[21–24] – a protocol which we will refer to as "projective". Other detection schemes in the hypothesis testing class have been advanced, most notably quantum illumination[25,26], ghost imaging – where the imaging

photons have not interacted with the imaged object[27–29], and imaging with undetected photons[30,31]. The interaction-free concept has touched off a flurry of research in the foundations of quantum mechanics, for example the Hardy paradox[32], non-local effects between distant atoms exchanging photons[33], and quantum engines[34].

Here we describe and demonstrate experimentally a hypothesis-testing protocol that employs repeated coherent interrogations instead of projective ones. In this protocol, the task is to detect the presence of a microwave pulse in a transmission line using a resonantly-activated detector realized as a transmon three-level device. We require that at the end of the protocol the detector has not irreversibly absorbed the pulse, as witnessed by a non-zero occupation of the second excited state. Clearly this task cannot be achieved with a classical absorption-based detector (e.g., a bolometer) or by using a simple two-level system as a detector. Our protocol is fundamentally different from the quantum Zeno interaction-free measurement: while in the latter case the mechanism of detection is the suppression of the coherent evolution by projection on the interferometer path that does not contain the object, in our protocol the evolution of the state of the superconducting circuit remains fully coherent. Surprisingly, this coherent addition of

[1]QTF Centre of Excellence, Department of Applied Physics, Aalto University, FI-00076 Aalto, Finland. ✉e-mail: shruti.dogra@aalto.fi; john.mccord@aalto.fi; sorin.paraoanu@aalto.fi

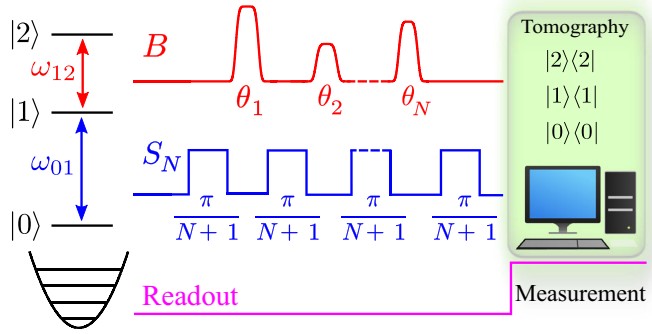

**Fig. 1 | Coherent interaction-free detection.** Schematic of the protocol, where $S_N$ and $B$ microwave pulses are shown in blue and red, respectively, along with the probe pulse for readout.

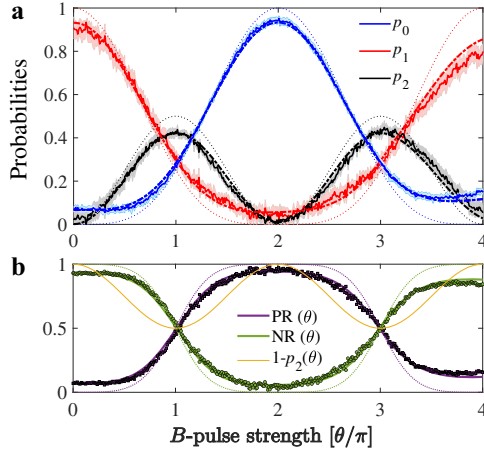

**Fig. 2 | Probabilities and associated positive/negative ratios for $N = 1$.**
**a** Probabilities vs. strength for a single $B$-pulse in our three-level system. The experimentally averaged profiles for the ground state ($p_0$), first excited state ($p_1$) and second excited states ($p_2$) are represented by blue, red and black colored continuous lines respectively. The corresponding colored dot-dashed lines are the simulated curves including decoherence and pulse imperfections, while the thin dotted lines show the ideal case. Each experimental curve is accompanied by a shaded region presenting the standard deviation of the mean obtained from 16 replicas of the same experiment. **b** Corresponding to each $B$-pulse strength, $PR(\theta) = p_0(\theta)/[p_0(\theta) + p_1(\theta)]$ and $NR(\theta) = p_1(\theta)/[p_0(\theta) + p_1(\theta)]$ obtained from the experiment are shown with purple circular markers and green square markers respectively, closely followed by the simulated purple and green continuous curves. The thin dotted lines represent the respective ideal cases, with no decoherence and without any experimental imperfections. The continuous yellow curve stands for the norm of the system subspace: $p_0(\theta) + p_1(\theta) = 1 - p_2(\theta)$.

amplitude probabilities results in a higher probability of successful detection.

This concept can be implemented in other experimental platforms where a three-level system is available. We note that projective interaction-free measurements have found already applications in optical imaging[35], counterfactual communication[36–41], ghost-imaging[42,43], detection of noise in mesoscopic physics[44], cryptographic key distribution[45,46], and measurement-driven engines[47]. We expect that our coherent version will be similarly adapted to these nascent fields.

In our experiments, we realize a series of $N$ Ramsey-like sequences by applying beam-splitter unitaries $S_N$ to the lowest two energy levels of a superconducting transmon. This creates the analog of the standard Mach-Zehnder spatial setup in a time-domain configuration[48]. The microwave pulses of strength $\theta_j$ that we wish to detect – which we will refer to as $B$-pulses – couple resonantly into the next higher transition, see Fig. 1. Specifically, let us denote the first three levels of the transmon by $|0\rangle$, $|1\rangle$, and $|2\rangle$ and the asymmetric Gell-Mann generators of SU(3) by $\sigma^y_{kl} = -i|k\rangle\langle l| + i|l\rangle\langle k|$, with $k, l \in \{0, 1, 2\}$. Microwave pulses applied resonantly to the 0–1 and 1–2 transitions respectively result in unitaries $S_N = \exp[-i\pi\sigma^y_{01}/2(N+1)]$ and $B(\theta_j) = \exp(-i\theta_j\sigma^y_{12}/2)$ (See Supplementary Information). The protocol employs a series of $j = \overline{1,N}$ Ramsey segments, each containing a $B$-pulse with arbitrary strength $\theta_j$, overall producing the evolution $U_N(\theta_1,...,\theta_N) = \prod_{j=1}^{N}[S_N B(\theta_{N+1-j})]S_N = S_N \prod_{j=1}^{N}[B(\theta_{N+1-j})S_N]$. Note that the absence of $B$-pulses results in $[S_N]^{N+1} = -i\sigma^y_{01} + |2\rangle\langle 2|$, acting nontrivially only on the subspace $|0\rangle, |1\rangle$ – therefore at the end of the sequence the entire ground-state population is transferred onto the first excited state $|0\rangle \to |1\rangle$. The goal is to ascertain the presence of $B$-pulses without absorbing them, that is, without creating excitations on level $|2\rangle$ of the transmon.

To understand the interaction-free physics in this setup, consider first a single sequence $N = 1$. The transmon is initialized in the ground state $|0\rangle$, which, when acted upon by $S_1$ ($\pi/2$ rotation around the $y$-axis in the $\{|0\rangle, |1\rangle\}$ subspace, corresponding to a 0.5 : 0.5 beam-splitter), drives the qubit into a coherent equal-weight superposition state $(|0\rangle + |1\rangle)/\sqrt{2}$. Next, the application of $B(\theta)$ (here we take $\theta_1 \equiv \theta$) and the subsequent application of $S_1$ results in the state $S_1 B(\theta) S_1 |0\rangle = \sin^2(\theta/4)|0\rangle + \cos^2(\theta/4)|1\rangle + (1/\sqrt{2})\sin(\theta/2)|2\rangle$, while if $B(\theta)$ is not present the final state is $|1\rangle$. By measuring dispersively the state of the transmon and finding it in the state $|0\rangle$, we can successfully ascertain the presence of the $B$ pulse without irreversibly absorbing it. On the other hand, if the transmon is found on $|1\rangle$ we cannot conclude anything, since this is also the result for the situation when the pulse is not present. For the ideal dissipationless case we have $p_0(\theta) = \sin^4(\theta/4)$, $p_1(\theta) = \cos^4(\theta/4)$ and $p_2(\theta) = (1/2)\sin^2(\theta/2)$. For $\theta = \pi$ this implies that we have $p_0(\pi) = 25\%$ chance of detecting the $B$-pulse without absorption, leaving $p_2(\pi) = 50\%$ as the probability of failure due to absorption.

Our protocol generalizes this concept to a series of $N \geq 1$ sequences, see Fig. 1, ending with detection by state tomography operators $D_0 = |0\rangle\langle 0|$, $D_1 = |1\rangle\langle 1|$, and $D_2 = |2\rangle\langle 2|$, which yield the success probability $p_0 = \langle D_0\rangle$, the probability of inconclusive results $p_1 = \langle D_1\rangle$, and the probability of absorption $p_2 = \langle D_2\rangle$. In addition, for a given string of $\theta_j$'s, as a key figure of merit we define the quantities relevant for the confusion matrix, as employed in standard predictive analytics. The Positive Ratio, $PR = p_0/[p_0 + p_1]$, is the fraction of cases where the interaction-free detection of $B$ is achieved *strictly speaking* without irreversible absorption. Its counterpart is the Negative Ratio, $NR = p_1/(p_0 + p_1)$, i.e., the fraction of experiments that are not accompanied by $B$ absorption, but for which we can not ascertain whether a $B$-pulse was present or not. In addition, the so-called interaction-free efficiency is sometimes utilized (see Supplementary Notes 1 and 2), which for the coherent case reads $\eta_c = p_0/(p_0 + p_2)$.

We obtain considerable enhancement of the success probabilities and efficiencies when detecting the pulses using this arrangement.

## Results
As described in the previous section, we use a transmon circuit with a dispersive readout scheme that allows us to measure simultaneously the probabilities $p_0$, $p_1$, and $p_2$. The 0–1 and 1–2 transitions are driven by two pulsed microwave fields, respectively implementing the $S_N$ unitaries and the $B$-pulses. Details of simulations and a description of the experimental setup are presented in Methods.

### Single $B$−pulse ($N = 1$)
The $N = 1$ case is important since it is the simplest realization of our concept, allowing us to present all the relevant experimental data and the most important figures of merit in a straightforward manner. The main results are shown in Fig. 2 and Fig. 3. Fig. 2a presents the probabilities $p_0$, $p_1$, and $p_2$ obtained experimentally, as well as a comparison with the simulated values and the ideal case. First, one notices that the

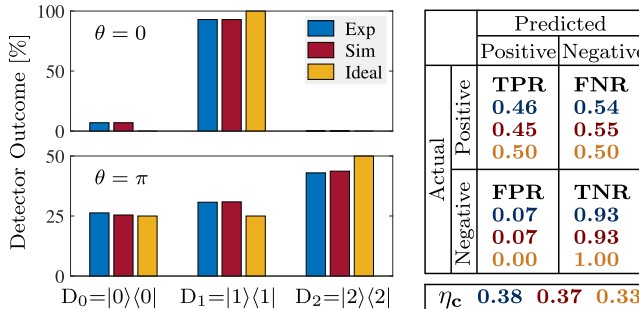

**Fig. 3 | Histogram of events for $\theta = \pi$ and $N = 1$, and the corresponding confusion matrix and efficiency.** (left panels) Histogram of events recorded by the detectors $D_0$, $D_1$, and $D_2$, which are modeled as projectors. Histograms resulting from the experiments, simulations and ideally expected values are shown in blue, red and yellow colors respectively. The results are obtained from $10^6$ realizations of the experiment, and for $B$-pulse strengths $\theta = 0, \pi$. The percentage outcome at $D_0$ corresponds to successful interaction-free detection, $D_2$ represents the number of times the pulse is absorbed, and $D_1$ are the inconclusive instances. (right panel) Confusion matrix and efficiency $\eta_c$ for the detection of $\pi$ pulses showing the experimental (blue), simulated (red) and ideally expected (yellow) values.

results are not invariant under $\theta \rightarrow \theta + 2\pi$, which is intrinsically related to the lack of invariance of spin-1/2 states under $2\pi$ rotations. Indeed, $B(\theta + 2\pi) = \exp(-i\pi\sigma_{12}^y)B(\theta)$ acts by changing the sign of the probability amplitudes on the subspace $\{|1\rangle, |2\rangle\}$, which subsequently alters the interference pattern after the second beam-splitter unitary. Then, we see that at $\theta = \pi, 3\pi$ the experimentally obtained probability for the interaction-free detection is 0.26; the same would also be expected in the projective case[17] (See Supplementary Information.).

From Fig. 2 we also notice that at $\theta = 2\pi$ the probability $p_0$ reaches a maximum (1 in the ideal case), while $p_1$ and $p_2$ are minimized (zero in the ideal case). This also happens if beam-splitters with $y$-axis rotation angles other than $\pi/2$ are used. It is a situation that has no classical analog: we are able to detect with near certainty a pulse that does not at all change the probabilities. As we will see next, when generalizing this result to $N > 1$ pulses, this maximum at $\theta = 2\pi$ extends to form a plateau of large $p_0$ values.

We can further characterize the detection capabilities of the $N = 1$ protocol by standard predictive analytics methods. In Fig. 3 we construct the histogram for the presence/absence of a $\theta = \pi$ $B$-pulse and we extract the associated confusion matrix by excluding the cases where the pulse is absorbed. The elements of the confusion matrix are defined by considering an actual positive or negative event (the pulse is either present or not present) and examining what can be predicted about the event based on the detector's response. Using standard terminology in hypothesis testing theory, for our device the elements of the confusion matrix are (see also Supplementary Table 1): when a $\pi$ $B$-pulse has actually been applied, we define the True Positive Ratio TPR $= p_0(\theta = \pi)/(p_0(\theta = \pi) + p_1(\theta = \pi)) = PR(\pi)$, which is the fraction of correct detections, and the False Negative Ratio FNR $= p_1(\theta = \pi)/(p_0(\theta = \pi) + p_1(\theta = \pi)) = NR(\pi)$, which is the fraction of inconclusive events. When the pulse is not applied, we have the False Positive Ratio FPR $= p_0(\theta = 0)/(p_0(\theta = 0) + p_1(\theta = 0)) = PR(0)$, which is the fraction of times we would wrongly predict that the pulse was applied, and its complementary True Negative Ratio TNR $= p_1(\theta = 0)/(p_0(\theta = 0) + p_1(\theta = 0)) = NR(0)$, which are the cases where we cannot predict anything. Finally, for the efficiency we obtain $\eta_c(\theta = \pi) = 0.33$ (refer to Supplementary Fig. 2 for other values). The experimental results in Fig. 3 are well reproduced by simulations and close enough to the ideal values.

## Two consecutive $B$-pulses ($N = 2$)

Next, we use our superconducting circuit to realize the coherent interaction-free detection of $N = 2$ pulses. The sequence of operations

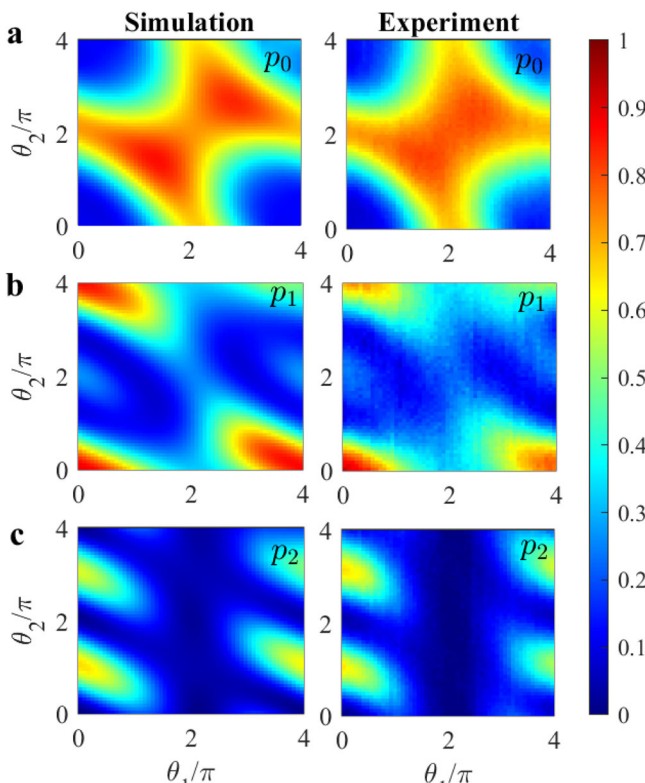

**Fig. 4 | Probabilities for the $N = 2$ case.** 2D probability maps for the **a** ground state ($p_0$), **b** first excited state ($p_1$), and **c** second excited state ($p_2$) as a function of $B$-pulse strengths $\theta_1$ and $\theta_2$.

consists of two independent $B$-pulses of strengths $\theta_1$ and $\theta_2$ sandwiched between three beam-splitter unitaries. In this case the coherent protocol already becomes fundamentally different from the projective one. Further, for $N = 2$, one can conveniently study all possible combinations of the pair of $B$-pulses whose strengths $\theta_1, \theta_2 \in [0, 4\pi]$ can be varied independently. This also allows us to study new situations, such as the absence of one of the $B$-pulses.

The experimental and the simulated results for the probabilities associated with the ground state, the first excited state and the second excited state as functions of $\theta_1$ and $\theta_2$ are shown in Fig. 4a–c, respectively. The Positive Ratio PR$(\theta_1, \theta_2) = p_0(\theta_1, \theta_2)/(p_0(\theta_1, \theta_2) + p_1(\theta_1, \theta_2))$ and the Negative Ratio NR$(\theta_1, \theta_2) = p_1(\theta_1, \theta_2)/(p_0(\theta_1, \theta_2) + p_1(\theta_1, \theta_2))$ as functions of $\theta_1$ and $\theta_2$ are shown in Fig. 5. Similar to the $N = 1$ case, the PR and NR can be used to construct the confusion matrix for any combination of $\theta_1$ and $\theta_2$ values. For the efficiency we obtain $\eta_c(\theta_1 = \pi, \theta_2 = \pi) = 0.81$ (refer to Supplementary Fig. 3 for other values). The experimental and simulated results are in very good agreement with each other, demonstrating control of the system over the full range of the two $\theta$-parameters.

To understand the difference between the coherent and the projective protocol, let us look at the case $\theta_1 = \theta_2 = \pi$. The projective protocol, if the first pulse is not absorbed, produces the state $|0\rangle$ at the input of the second beam-splitter unitary (see Supplementary Note 2). As a result, the second Ramsey sequence provides another round of monitoring the pulse, though this is essentially only a repetition of the first. In contrast, in the coherent protocol the input to the second beam-splitter unitary is a superposition of $|0\rangle$ and $|2\rangle$. The second monitoring of the pulse retains the amplitude of $|2\rangle$ in a coherent way, resulting in a higher probability of success. This unexpected effect can be seen by a straightforward calculation for the ideal case and $\theta_1 = \theta_2 = \pi$, which yields probabilities $p_0 = 0.8091$, $p_1 = 0.0034$, $p_2 = 0.1875$, and PR $= 0.99$; whereas, the equivalent respective figures for the projective case are 0.4219, 0.1406, 0.4375, and 0.75.

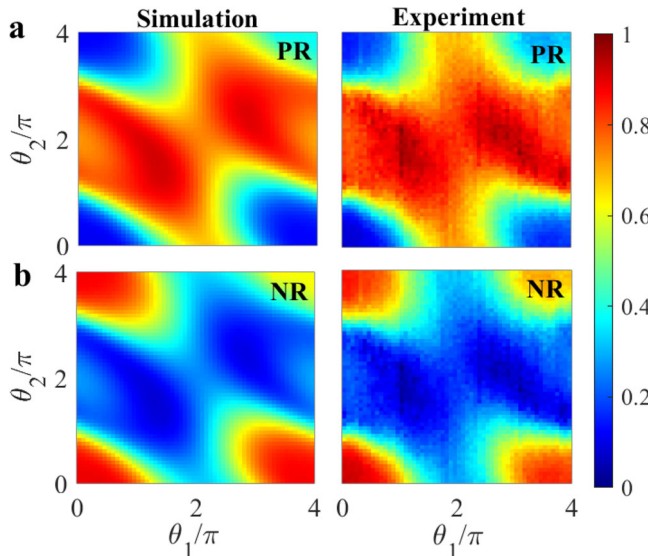

**Fig. 5 | Positive and negative ratios for the $N = 2$ case.** Simulated and experimental 2D maps for the **a** Positive Ratio PR($\theta_1, \theta_2$) and for the **b** Negative Ratio NR($\theta_1, \theta_2$) as a function of $\theta_1$ and $\theta_2$.

## Multiple consecutive *B*-pulses ($N > 2$)

Next, we use our superconducting circuit to realize the coherent interaction-free detection of $N > 2$ pulses, where we observe even more efficient coherent accumulation of the amplitude probabilities on the state $|0\rangle$ under successive interactions with the *B*-pulse and applications of Ramsey $S_N$ (See Supplementary Information).

In these experiments we use both equal-strength pulses $\theta_j = \theta$ and pulses with randomly-chosen $\theta_j \in \{0, \pi\}, j = \overline{1,N}$, while the beam-splitter unitary is a $\pi/(N+1)$ rotation around the $y$ axis in the $\{|0\rangle, |1\rangle\}$ subspace. To recall, in the absence of the *B*-pulses we have $[S_N]^{N+1}$ and in the presence of the *B*-pulses we have $S_N \prod_{j=1}^{N} [B(\theta_{N+1-j}) S_N]$. The results are presented in Fig. 6. Due to the multidimensional nature of these experiments we focus here on $p_0$; other possible figures of merit are presented in Supplementary Information Note 2(c).

The large-$N$ experimental sequences have a significant time cost with the worst case of 25 *B*-pulses corresponding to 4.3 μs, which is even longer than the relaxation time $\Gamma_{10}^{-1} = 3.4$ μs (see Methods for details). Thus, in addition to the standard three-level Lindblad master equation[49,50], in order to accurately model the system we may include a depolarizing channel $\rho(t) \rightarrow (1 - \epsilon)\rho(t) + \epsilon \mathbb{I}_3/3$[51] (see Methods). Here we assume that the imperfections in the $1 - 2$ drive results in mixing of the qutrit state; hence the parameter $\epsilon$ is taken as directly proportional to the pulse amplitude, given by $\epsilon[\theta] = 1.8 \times 10^{-3} \times \theta/\pi$. This choice of model fits our experimental data very well as shown in Fig. 6, where continuous lines correspond to the simulation including the depolarizing channel and dotted lines correspond to the simulation without the depolarizing channel. As expected, the overall effect of depolarization is more prominent for a larger number of *B*-pulses and for large $\theta$. In all of these plots, experimental results are shown by markers with experimental error bars (standard deviation about the mean by four repetitions of the same experiment). Small deviations of the experimental values from the ideal results are due to decoherence and pulse errors. Larger values of $p_0$ correspond to a higher probability of interaction-free detection. We have verified numerically that with increasing $N$, $p_0$ increases, approaching 1 in an ideal case.

In the case of equal-strength pulses, for each $N$, we perform a total of $\mathcal{M}$ experiments, with the *B*-pulse strength varying linearly with the experiment number as: $\theta = \theta_{j,m} = m\pi/\mathcal{M}$ with labels: $j = \overline{1,N}$ and $m = \overline{1,\mathcal{M}}$ such that $\theta \in [0, \pi]$. The results for the overall success probability $p_0$ are shown in Fig. 6a, for various numbers $N \in [1,25]$ of *B*-

pulses and $\mathcal{M} = 180$. Simulated and experimental $p_0$ values are shown as surface plots in parts (i) and (ii) respectively.

Interestingly, with increasing number of *B*-pulses, the final $p_0$ is independent of the *B*-pulse strength ($\theta$), and has a tendency to reach large values. As anticipated, a plateau characterized by high values is formed, which is the extension to smaller $\theta$'s of the maximum seen in the $N = 1$ case around $\theta = 2\pi$. This is also clearly reflected from the plot in Fig. 6a(iii) showing the mean value of $p_0$ (E[$p_0$] in red) resulting from experiments with different *B*-pulse strengths versus the number of Ramsey sequences. The 'no *B*-pulse' situation is shown with black square markers and that of maximum *B*-pulse strength is shown with blue triangular markers, where the increase in $p_0(\theta = 0)$ with $N$ and lower values of $p_0(\theta = \pi)$ is due to the decoherence. It is clear from the three curves that E[$p_0$] tends to approach the higher limiting values, which is attributed to the larger plateau of high $p_0$ values with increasing $N$ (see Supplementary Figs. 6 and 7). As a direct consequence of the plateau formation, the minimum value of $\theta$ that gives rise to near maximal $p_0$ is much smaller than $\pi$ for large $N$. The standard deviation of the $p_0$ distribution versus $N$ is shown in Fig. 6a(iv). Each of these experimental values are accompanied by simulations, demonstrating quite close agreement. A comparison (see Supplementary Notes 2 and 3) with the projective case - for which exact analytical results are available - demonstrates the advantage of the coherent protocol for all values of $N$.

We also study the case of randomly-chosen $\theta_j \in \{0, \pi\}, j = \overline{1,N}$, with results shown in Fig. 6b. Panels (i), (ii) present surface maps of the simulated and experimental $p_0$ versus $N$ and $m$, where $\mathcal{M} = 400$. Experimental and simulated mean- E[$p_0$], minimum- $p_0^{(\min)}$, and maximum- $p_0^{(\max)}$ values obtained from this distribution are shown in panel (iii) with markers and continuous curves respectively. The standard deviation $\sigma[p_0] = \sqrt{E[(p_0 - E[p_0])^2]}$ of $p_0$ versus $N$ is shown in part (iv). Again, we observe that the mean value of $p_0$ increases with $N$, while the standard deviation of repeated measurements decreases with N. Thus, for a large N, the *B*-pulse strength does not matter anymore, and we obtain a highly effective interaction-free detection. Surprisingly, the case with random *B*-pulse strengths appears to outperform the case with identical *B*-pulses. Comparing parts a(iii) and b(iii) of Fig. 6, the success probability of the coherent interaction-free detection in the worst case (green curve) for random *B*-pulse strengths is already high enough, with a maximum value (for $N = 25$) of $0.83 \pm 0.03$ (experiment) and 0.82 (simulation), close to the mean values E[$p_0$] = $0.88 \pm 0.03$ (experimental) and E[$p_0$] = 0.87 (simulated). On the other hand, in the case of identical *B*-pulses, the mean values for $N = 25$ are only E[$p_0$] = $0.81 \pm 0.01$ (experiment) and E[$p_0$] = 0.80 (simulation), even slightly below the worst-case scenario with random pulses. Also, especially at large $N$'s, the standard deviation about the mean value of the distribution is much lower in the case of random *B*-pulses as opposed to the identical *B*-pulses case, which is clear upon comparison of Fig. 6a(iv) and b(iv). Thus, an adversarial attempt to randomize the *B*-pulse strengths in order to evade detection has, surprisingly, the opposite effect, improving the interaction-free coherent detection.

In Fig. 6c we provide a histogram representation of the $p_0$ distributions for $N = 5,15,25$. The distribution in red in all three cases corresponds to $\theta_j = \theta = 0$ - and hence lie at the lower limit of $p_0$ range, while the distribution in yellow represents the case $\theta_j = \theta = \pi$ and lies close to the upper limit. The interesting part is the distribution in blue with arbitrarily chosen *B*-pulse strengths $\theta_j = \theta \in [0, \pi]$, which moves towards the right side and tends to squeeze with increasing $N$. The same idea is conveyed by the increasing mean value (E[$p_0$]) and decreasing standard deviation with $N$ as discussed earlier.

Finally, as another figure of merit for the protocol, we can obtain PR($\theta$) and NR($\theta$) for *B*-pulses with equal strengths $\theta_{j,m} = \theta \in [0, \pi]$ for each $N \in [1,25]$. The detailed surface maps presenting the ideal case (without decoherence), and the simulated and experimentally obtained values for PR($\theta$) and NR($\theta$) at various $N$ are shown in Supplementary

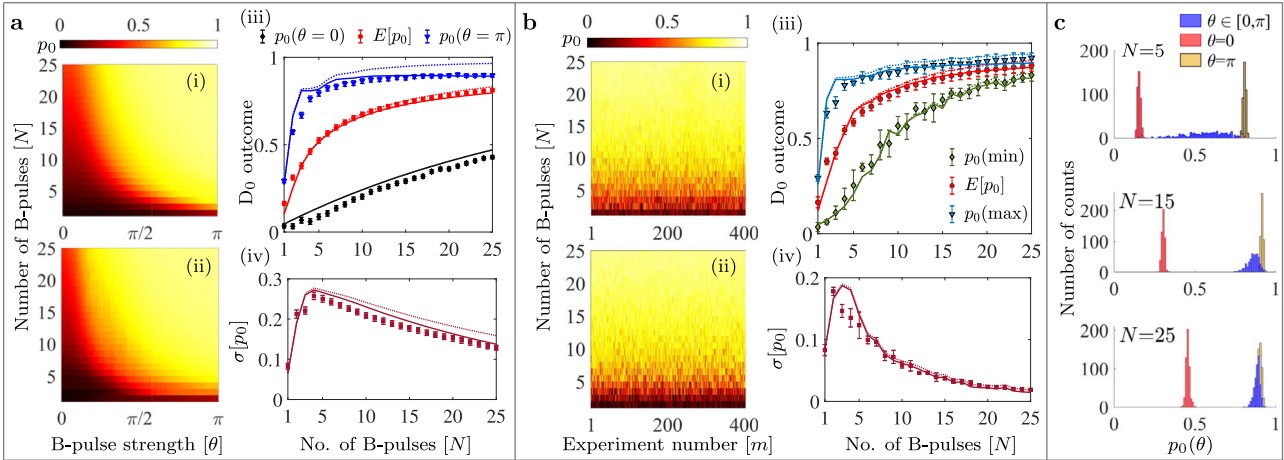

**Fig. 6 | Results for the $D_0$ outcome in the case of multiple Ramsey sequences with $B$-pulses $N \in [1\,25]$ of varying strengths, $\theta_{j,m} \in [0, \pi]$ with $j = \overline{1,N}$ and $m$ indexing the experimental realization for a given $N$. a** Plots for identical $B$-pulses $\theta_{j,m} = \theta = m\pi/\mathcal{M}$ for a given $N$: (i) Simulated and (ii) experimentally obtained maps. (iii) Values of $p_0$ at $B$-pulse strengths $\theta = 0, \pi$ and mean $p_0$ (E[$p_0$]) versus $N$ with markers with error bars showing the experimental results and the corresponding continuous lines obtained from the simulations. Red circular markers present the mean value $E[p_0]$, black diamond markers correspond to the case with no $B$-pulses ($\theta = 0$) and data points with blue triangular markers stand for the case of maximum $B$-pulse strengths ($\theta = \pi$). (iv) Standard deviation evaluated for a given $N$ versus $N$. **b** Plots for arbitrarily chosen $\theta_{j,m} \in [0, \pi]$: (i) Simulated and (ii) experimental data for $p_0$ as a function of $N$ and $m$. (iii) Simulated (continuous lines) and experimental

(markers with error bars) of mean ($E[p_0]$ in red) and extremum values ($p_0$(min) in green and $p_0$(max) in blue). (iv) Standard deviation of $p_0$ versus $N$. Dotted curves in all the plots are simulations without the inclusion of depolarization. Error bars in all the plots correspond to standard deviation of the measured quantities with respect to their respective mean values, obtained from four repetitions of the full experiment. **c** Histogram of the experimental $D_0$ counts for various system sizes ($N = 5, 15, 25$) with $B$-pulses of arbitrary strengths, $\theta \in [0, \pi]$ (in blue) compared with those of $B$-pulse strengths: $\theta = \pi$ (in yellow) and $\theta = 0$ (no $B$-pulse) in red. Clearly as the system size increases, the strengths of the $B$-pulses become less significant and approach the clustering near the maximal $p_0$, which is a signature of the highly efficient interaction-free detection.

Fig. 5. Similar to the previous cases, these can be used to define the elements of the confusion matrix, for example TPR = PR($\pi$), FPR = PR(0), etc. We find that at large $N$ the positive ratio reaches high values for a wide range of $\theta$'s, altogether forming a plateau of stable and high-confidence interaction-free detection. Correspondingly, a wide region of low NR($\theta$) values are obtained. For example, from the experimental data, for $N = 5, 15, 25$ the value PR($\theta$) = 0.90 is reached at $\theta = 0.54\pi, 0.32\pi, 0.18\pi$ respectively, going up to $\approx 0.95$ at $\theta = \pi$. The corresponding values of the efficiency $\eta_c$ for the same $N$ and $\theta$ combinations are 0.67, 0.81 and 0.81 respectively, see also Supplementary Fig. 6.

## Discussion

In our protocol quantum coherence serves as a resource, yielding a significantly high detection success probability. The enhancement can be understood as the coherent accumulation of amplitude probabilities on the state $|0\rangle$ under successive interactions with the $B$-pulse and applications of Ramsey $S_N$ (see Supplementary Note 3), by making use of the full 3−dimensional Hilbert space at each step. In contrast, the projective protocol[21,22] employs the quantum Zeno effect to confine the dynamics in the $|0\rangle, |1\rangle$ subspace after each interaction with the pulse. Thus, it extracts which-way information about the presence or absence of the pulse at each step of the protocol.

To gain more insight into the functioning of our protocol, consider the case of uniform $B\pi$-pulses. We have verified numerically that at large values of $N$ the following approximate relation holds

$$U_N(\theta_1 = \pi, ..., \theta_N = \pi) = [S_N B(\pi)]^N S_N \overset{N \gg 1}{\approx} |0\rangle\langle 0| + \left(-i\sigma_{12}^y\right)^N$$

We can also provide a consistency argument for this relation: since we are dealing with $\pi$ pulses only, we have $B(\pi) = |0\rangle\langle 0| - i\sigma_{12}^y$, and since $N \gg 1$ we can write also $S_{N+1} \approx \mathbb{I}_3$. Then, assuming the above expression, we can estimate $U_{N+1}(\theta_1 = \pi, ..., \theta_{N+1} = \pi) \approx U_N(\theta_1 = \pi, ..., \theta_N = \pi)B(\pi)\mathbb{I}_3 = |0\rangle\langle 0| + \left(-i\sigma_{12}^y\right)^{N+1}$. Thus, if we start from the ground state, the dynamics tend to stabilize this state at large $N$, which results in

the appearance of plateaus of near-unity $p_0$ in Fig. 6 a. This is in some sense the closest counterpart of the approximation $\left[\cos(\pi/2(N+1))\right]^{2(N+1)} \overset{N \gg 1}{\approx} 1$, which is crucial for establishing a large detection in the standard projective case (see also Supplementary Note 2).

In the experimental realization of projective interaction-free measurements, as done with bulk optics[22] or waveguide circuits[23], the maximum experimental efficiencies obtained are 0.73 and 0.63 resepectively, both obtained for $N = 9$. For larger $N$'s it is observed that the efficiency decreases due to losses. By contrast, in our case the efficiency for $N = 9$ is $\eta_c(\theta = \pi) = 0.89$ and it increases further as $N$ gets larger, reaching 0.96 at $N = 20$ (see also Supplementary Fig. 6). Our protocol also compares favorably with other realizations of microwave photon detection, based for example on Raman processes[52], or on cavity-assisted conditional gates[53,54]. The dark count rate, which is the number of counts per unit time in the absence of a pulse, can be obtained from FPR $\approx p_0(\theta = 0)$ divided by the sensing time: we obtain 0.1 counts/$\mu$s. This can be further improved without affecting the true positives by reducing the decoherence and the effective qubit temperature at the beginning of the protocol, for example by using active reset. The experimentally-demonstrated detection bandwidth of our system is given by the inverse minimum duration of the $B$-pulses used in the experiment; e.g., for the 56 ns pulses this corresponds to a 18 MHz bandwidth.

The coherent interaction-free protocol can also be represented geometrically on the unit 2−sphere. In the Majorana representation[55], a three-level system is represented by two points $\mathcal{S}_1(x,y,z)$ and $\mathcal{S}_2(x,y,z)$ – called Majorana stars – on the surface of this sphere [56]. In our protocol, the system is initialized in the state $|0\rangle$, which corresponds to both Majorana stars residing at the North Pole, $\mathcal{S}_{1,2}^i(0,0,0)$. In the absence of $B$-pulses, the protocol ends with one star at the North Pole and the other at the South Pole. In the presence of $B$-pulses with $\theta_j = \pi$, we find that both stars are located in the northern hemisphere for N≥2, and they tend to get closer and closer to the North Pole with increasing $N$ (see also Supplementary Note 6). To illustrate this, in Fig. 7a−c we

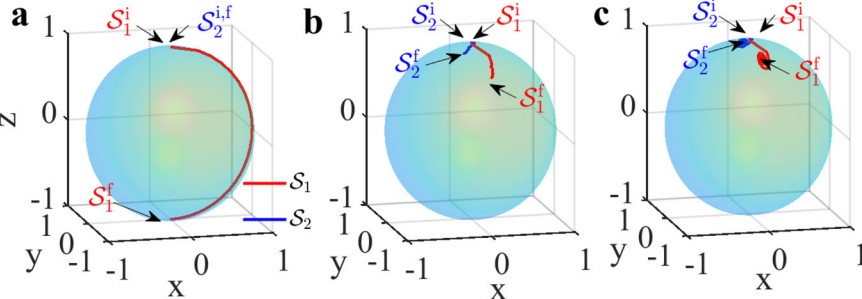

**Fig. 7 | Majorana representation.** Averaged Majorana trajectories followed by the three-level system for $N = 25$ in the case of **a** no $B$-pulse, $B$-pulses with **b** equal strengths, and **c** with randomly chosen strengths in the range $[0,\pi]$. Trajectories of the Majorana stars $\mathcal{S}_1$ and $\mathcal{S}_2$ are shown in red and blue colors respectively, where $\mathcal{S}_{1,2}^{i}$ marks the initial state and $\mathcal{S}_{1,2}^{f}$ correspond to the final state of the three-level system on the Majorana sphere.

present the resulting trajectories of the Majorana stars ($\mathcal{S}_1$ in red and $\mathcal{S}_2$ in blue) for the case of no $B$-pulse, $B$-pulses with equal strengths, and $B$-pulses with randomly chosen strengths respectively. Here we took $N = 25$, such that each Majorana trajectory consists of 26 points; the initial and final stars of the trajectories are labelled as $\mathcal{S}_{1,2}^{i}$ and $\mathcal{S}_{1,2}^{f}$ respectively. The trajectories correspond to the average states obtained from 400 repetitions of the protocol with varying $B$-pulse strengths (as discussed in the previous section). The presence of both Majorana stars in the vicinity of the North Pole on the sphere serves as a sensitive geometrical signature of the interaction-free detection of the $B$-pulses. There is a clear difference between the situation of no $B$-pulse, where one Majorana star is at the North Pole $(0,0,1)$ and the other at the South Pole $(0,0,-1)$, as compared to the presence of the $B$-pulse, shown in Fig. 7b and c, where both $\mathcal{S}_1$ and $\mathcal{S}_2$ end up close to the North Pole. Comparing Fig. 7b, c, we find that the $z$-coordinates of the final Majorana stars in the case of equal $B$-pulse strengths is 0.7381, while the minimum value of the $z$-coordinate reached in the case of randomly chosen $B$-pulse strengths is 0.7863. Clearly, in the case of randomly chosen $B$-pulse strengths the respective Majorana trajectories are confined closer to the North Pole, confirming the results from the previous section.

We point out that these results can be extended in various directions. For example, they can be applied for the non-invasive monitoring of microwave currents and pulses, which is an open problem in quantum simulation[57]. They provide a proof of concept for a photon detector, conceptually and practically different from realizations based on other principles, that can be further optimized. Our protocol works also when the $B$-pulse is a Fock state and it can be utilized to assess non-destructively the presence of photons stored in superconducting cavities (see Supplementary Note 2). This can be utilized for axion detection, where the generation of a photon is expected to be a rare event. Here also the existing detectors have a high dark count rate; thus, one can increase the confidence level by assessing its presence first non-destructively and then confirming it by more conventional means.

In conclusion, we proposed a coherent interaction-free process for the detection of microwave pulses and we realized it experimentally with a superconducting quantum circuit. For the case of a single pulse with strength $\theta = \pi$, we obtain an interaction-free detection probability of 0.26. Further, we emulated multiple Ramsey sequences and we obtained a highly efficient interaction-free detection of the $B$-pulse. We observed that for a large number of sequences a detection probability approaching unity is obtained irrespective to the strength of the pulses, and, surprisingly, this probability is even higher when the pulses have random strength.

## Methods
### Experimental setup
A schematic of the setup is shown in Fig. 8. The sample is mounted in a dilution refrigerator via a sample holder which is thermally

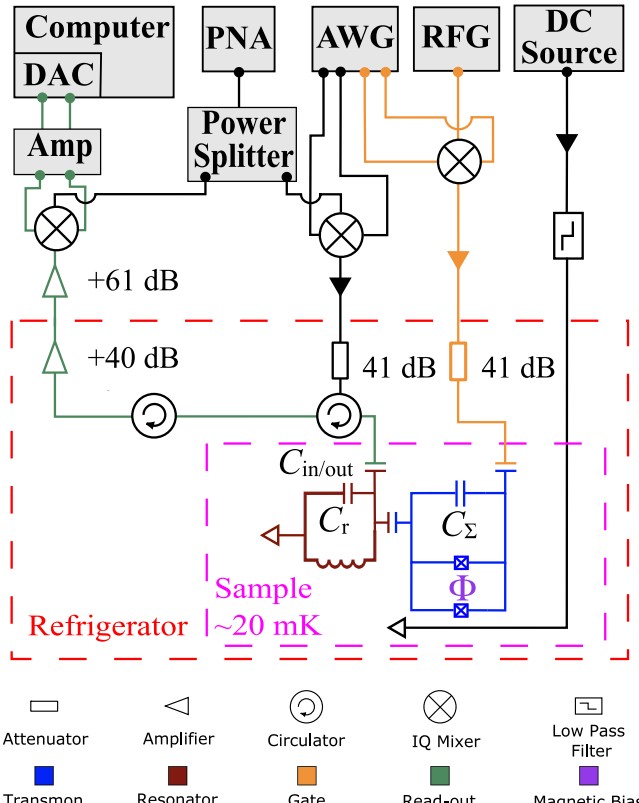

**Fig. 8 | Experimental setup.** Schematic of the experimental setup used in this work, including the transmon circuit's integration with the dilution refrigerator and microwave electronics.

anchored to the mixing chamber. There are several lines that connect our sample to the external circuitry: the microwave gate line which delivers the microwave drive pulses to the transmon, a flux-bias line which provides a constant DC magnetic field, and the measurement line which is capacitively coupled to the readout resonator via an input/output capacitor. The flux-bias line sends a current near the SQUID loop, which induces a magnetic flux and thus enables the transmon transition frequency to be tuned. To reduce the sensitivity of the device to charge noise, the SQUID loop is shunted by a large capacitance[58–60] denoted by $C_\Sigma$ in Fig. 8. The transmission line is used to probe the resonator by sending microwave pulses or continuous signals into it.

The drive pulses used to realize the beam-splitter unitaries and the $B$-pulses have super-Gaussian envelopes ($\propto \exp[-(t/\tau)^4/2]$) with the

following time-dependence:

$$\Omega(t) = \Omega_0 \exp\left[-\frac{1}{2}\left(\frac{t}{\tau}\right)^4\right] \quad (1)$$

where $\Omega_0^{(S_N)} = \pi/[(N+1)\int_{-\tau_c}^{\tau_c}\exp[-(t/\tau)^4/2]dt]$ for beam-splitters and $\Omega_0(\theta) = \theta/\int_{-\tau_c}^{\tau_c}\exp[-(t/\tau)^4/2]dt$ for the $B$-pulses. Thus, the effective pulse area is determined by $\int_{-\tau_c}^{\tau_c}\exp[-(t/\tau)^4/2]dt$, where $\pm\tau_c$ are the start and the end points of the drive pulse (the points where the pulse is truncated) and $\tau$ is a time constant. In our experiments $\tau = 14$ ns and $\tau_c = 2\tau = 28$ ns, which corresponds to a total pulse length of 56 ns and an effective pulse area $\int_{-\tau_c}^{\tau_c}\exp[-(t/\tau)^4/2]dt = 30.18$ ns. The amplitude $\Omega_0$ is determined from Rabi oscillations measurements varying the amplitude of the transmon drive pulse and its frequency while keeping the pulse duration fixed. The variation of the pulse amplitude is achieved using I and Q waveform amplitudes from our arbitrary waveform generator (AWG), which are mixed in an IQ mixer with the LO tone generated by a continous microwave generator (AWG). We utilize a homodyne detection scheme for determining the state of the transmon. A microwave source (PNA) provides a continuous signal at the LO frequency for our readout pulse as well as that for the demodulated reflected signal from the resonator. As such, a power splitter is employed to halve this signal, where one part is sent to the LO port of an IQ mixer which modulates a probe pulse with readout rectangular envelopes from the I and Q quadratures generated by the AWG. The other part is sent to an IQ mixer which demodulates the signal reflected back from the resonator. After demodulation the quadratures of this mixer are amplified and subsequently digitized and recorded via our data acquisition card (DAC).

## Decoherence model and numerical simulations

In the rotating wave approximation (RWA), the transmon Hamiltonian in the three-level truncation is

$$
\begin{aligned}
H(t) = &\frac{\hbar}{2}\left[\Omega_{01}(t)e^{i\phi_{01}}|0\rangle\langle 1| + \Omega_{01}(t)e^{-i\phi_{01}}|1\rangle\langle 0| + 2\delta_{01}|1\rangle\langle 1|\right] \\
&+ \frac{\hbar}{2}\left[\Omega_{12}(t)e^{i\phi_{12}}|1\rangle\langle 2| + \Omega_{12}(t)e^{-i\phi_{12}}|2\rangle\langle 1| + 2(\delta_{01}+\delta_{12})|2\rangle\langle 2|\right]
\end{aligned}
\quad (2)
$$

where the drive amplitudes follow the form as per Eq. (1), and are denoted as $\Omega_{01}(t)$ and $\Omega_{12}(t)$ for the $|0\rangle - |1\rangle$ and $|1\rangle - |2\rangle$ transitions respectively, carrying the respective phase factors $e^{\pm i\phi_{01}}$ and $e^{\pm i\phi_{12}}$[50]. With the notation $\sigma_{kl} = |k\rangle\langle l|$, and assuming resonance $\delta_{01} = \delta_{12} = 0$, the Hamiltonian reads

$$H(t) = \frac{\hbar\Omega_{01}(t)}{2}e^{i\phi_{01}}\sigma_{01} + \frac{\hbar\Omega_{12}(t)}{2}e^{i\phi_{12}}\sigma_{12} + \text{h.c.} \quad (3)$$

To introduce dissipation, we use the standard Lindblad master equation, where $D[L]\rho = L\rho L^\dagger - \frac{1}{2}\{L^\dagger L, \rho\}$ is the Lindblad super operator and $L$ is the jump operator applied to the density matrix $\rho$. For our three-level system we have (see e.g.[61,62])

$$\dot{\rho} = -\frac{i}{\hbar}[H,\rho] + \sum_{k,l=0,1,2}^{k\neq l}\Gamma_{k\to l}D[\sigma_{lk}]\rho + \sum_{k=0,1,2}\frac{\Gamma_k^\phi}{2}D[\sigma_{kk}]\rho,$$

where $\Gamma_{k\to l}$ is the excitation/decay rate between states $|k\rangle$ and $|l\rangle$, and $\Gamma_k^\phi$ is the dephasing rate associated with level $k$. The operators $\sigma_{lk} = |l\rangle\langle k|$ with $k > l$ are lowering operators and those with $k < l$ are raising operators corresponding to the transition $lk$. The Lindblad dephasing operators act only on the off-diagonal matrix elements, while the relaxation operators act on both the diagonal and off-diagonal matrix elements. However, since we operate on transitions, the individual dephasing rates $\Gamma_k^\phi$ cannot be determined directly from experiments. Instead, we can rewrite the equation above in a form that

involves only pairs of levels[50]

$$
\begin{aligned}
\dot{\rho} = &-\frac{i}{\hbar}[H,\rho] + \Gamma_{2\to 1}\rho_{22}(\sigma_{11} - \sigma_{22}) + \Gamma_{1\to 0}\rho_{11}(\sigma_{00} - \sigma_{11}) \\
&+ \Gamma_{1\to 2}\rho_{11}(\sigma_{22} - \sigma_{11}) + \Gamma_{0\to 1}\rho_{00}(\sigma_{11} - \sigma_{00}) \\
&- \sum_{k,l=0,1,2}^{k\neq l}\gamma_{kl}\rho_{kl}\sigma_{kl},
\end{aligned}
$$

where the relaxation rates satisfy the detailed balance condition $\Gamma_{k\to l} = e^{-\hbar\omega_{kl}/k_B T}\Gamma_{l\to k}$ (with $l > k$) at a temperature T with $k_B$ being the Boltzmann constant and $\hbar\omega_{kl}$ being the energy level spacing between the $k^{th}$ and $l^{th}$ levels. By introducing the occupation numbers $n_{kl} = 1/[\exp(-\hbar\omega_{kl}/k_B T) - 1]$, the rates $\Gamma_{k\to l}$ can be expressed in terms of the zero-temperature decay rates $\Gamma_{lk}$ (with $l > k$) as $\Gamma_{k\to l} = n_{kl}\Gamma_{lk}$ ($l > k$) and $\Gamma_{l\to k} = (n_{kl} + 1)\Gamma_{lk}$ ($l > k$). It is clear from this decoherence model that the relaxation rates $\Gamma_{k\to l}$ for $k < l$ are significant only at higher temperatures of several tens of mK, which lead to transitions from lower to higher energy levels. The decay rates for the off-diagonal matrix elements are $\gamma_{10} = \gamma_{01} = (\Gamma_{1\to 0} + \Gamma_{0\to 1})/2 + \Gamma_{10}^\phi$, $\gamma_{21} = \gamma_{12} = (\Gamma_{1\to 2} + \Gamma_{2\to 1})/2 + \Gamma_{21}^\phi$, and $\gamma_{20} = \gamma_{02} = (\Gamma_{1\to 0} + \Gamma_{2\to 1} + \Gamma_{0\to 1} + \Gamma_{1\to 2})/2 + \Gamma_{20}^\phi$. Here we define the dephasing rates associated with each transition as $\Gamma_{kl}^\phi = \Gamma_{lk}^\phi = (\Gamma_k^\phi + \Gamma_l^\phi)/2$. Note that the off-diagonal decay of the matrix elements $\rho_{kl}$ due to dephasing can be understood as resulting from $\mathbb{I}_{kl}D[\sigma_{kl}^z]\rho_{kl} = \sigma_{kl}^z\rho\sigma_{kl}^z - \mathbb{I}_{kl}\rho\mathbb{I}_{kl}$, which is the familiar qubit dephasing expression projected onto the $\{|k\rangle, |l\rangle\}$ subspace, with $\sigma_{kl}^z = \sigma_{kk} - \sigma_{ll} = |k\rangle\langle k| - |l\rangle\langle l|$ and $\mathbb{I}_{kl} = \sigma_{kk} + \sigma_{ll} = |k\rangle\langle k| + |l\rangle\langle l|$.

## Experimental parameters and sample specifications

For the $N = 1$ and $N = 2$ cases, experiments have been performed on a sample with $|0\rangle - |1\rangle$ and $|1\rangle - |2\rangle$ transition frequencies $\omega_{01}/(2\pi) = 5.01$ GHz and $\omega_{12}/(2\pi) = 4.65$ GHz. The simulations make use of the general form of the Lindblad master equation for the quantum state evolution with relaxation and dephasing rates obtained from standard characterization measurements: $\Gamma_{10} = 0.72$ MHz, $\Gamma_{21} = 1.55$ MHz, $\Gamma_{10}^\phi = 0.4$ MHz, $\Gamma_{21}^\phi = 0.6$ MHz, and $\Gamma_{02}^\phi = 1$ MHz. The duration of the beam-splitter pulse is 56 ns (see also Eq. (1)) and the amplitude of the pulse is directly proportional to the angle of rotation (in a given subspace). The $B$-pulses however have a fixed duration of 56 ns until $\theta = 3.38\pi$, beyond which the upper limit of the output power from our arbitrary waveform generator (AWG) is reached. To tackle this issue, the pulse duration is gradually increased from 56 ns to 61 ns in steps of 1 ns (as $\theta$ varies from $3.38\pi$ to $4\pi$), such that the desired pulse-area is attained with lower pulse amplitudes. The transmon starts in thermal equilibrium at an effective temperature of 50 mK (measured independently, see[63]) such that the initial probability of occupation of the ground state, first excited state and second excited state is $p_0 = 0.9917 = 99.17\%$, $p_1 = 0.0082 = 0.82\%$, and $p_2 = 0.0001 = 0.1\%$.

For experiments involving a large number of pulses ($N > 2$) we use a sample with $\omega_{01}/(2\pi) = 7.20$ GHz and $\omega_{12}/(2\pi) = 6.85$ GHz. The relaxation and dephasing rates obtained from independent measurements are $\Gamma_{10} = 0.29$ MHz, $\Gamma_{21} = 1.15$ MHz, $\Gamma_{10}^\phi = 0.18$ MHz, $\Gamma_{21}^\phi = 1.82$ MHz, and $\Gamma_{02}^\phi = 1.70$ MHz. All the beam-splitter pulses are 56 ns and $B$-pulses are of duration 112 ns with various different amplitudes. For the case of identical $B$-pulses, $\theta$ is increased linearly from 0 to $\pi$ in 180 steps and in each case $p_0$ is measured for $N \in [1, 25]$. To obtain the error bars, each experiment is repeated four times. In the case of random $B$-pulses, random strengths are chosen arbitrarily from a uniform distribution of random numbers from 0 to $\pi$. Error bars result from the four repetitions of the same experiment. The corresponding surface maps, histograms and mean and standard deviation values are presented and discussed in the main text. For further details on the errors due to pulse imperfections, see Supplementary Note 5.

For very long experiments, it is known that we can accumulate errors resulting in excess populations on the higher energy levels. The

standard description for this effect is via an additional depolarizing channel[51]. For a three-level system the depolarizing channel can be written in the operator-sum representation[64], which is a completely positive trace-preserving map, such that the final state is given by

$$\rho_f = \sum_\nu K_\nu \rho K_\nu^\dagger, \quad \text{with} \quad \sum_\nu K_\nu^\dagger K_\nu = \mathbb{I}_3. \tag{4}$$

The Kraus operators $K_\nu$'s are given in terms of Gell-Mann matrices: $K_1 = \sqrt{\epsilon/6}\lambda_1$, $K_2 = \sqrt{\epsilon/6}\lambda_2$, $K_3 = \sqrt{\epsilon/6}\lambda_4$, $K_4 = \sqrt{\epsilon/6}\lambda_5$, $K_5 = \sqrt{\epsilon/6}\lambda_6$, $K_6 = \sqrt{\epsilon/6}\lambda_7$, $K_7 = \sqrt{\epsilon/3}\lambda_3$, $K_8 = \sqrt{\epsilon/6}(\sqrt{3}\lambda_8 - \lambda_3)$, $K_9 = \sqrt{\epsilon/6}(\sqrt{3}\lambda_8 + \lambda_3)$, and $K_{10} = \sqrt{1 - 8\epsilon/9}\,\mathbb{I}_3$. Here, $\lambda_{1(2)} = \sigma_{01}^{x(y)}$, $\lambda_{4(5)} = \sigma_{02}^{x(y)}$, $\lambda_{6(7)} = \sigma_{12}^{x(y)}$, $\lambda_3 = \sigma_{01}^z$, and $\lambda_8 = (\sigma_{02}^z + \sigma_{12}^z)/\sqrt{3}$. The final state following Eq. (4) is

$$\rho_f = \frac{\epsilon \mathbb{I}_3}{3} + (1 - \epsilon)\rho . \tag{5}$$

In other words the system is replaced with the completely mixed state $\mathbb{I}_3/3$ with probability $\epsilon$ – otherwise it is unaffected, with probability $1 - \epsilon$. We consider only the depolarization caused by the $B$-pulse, with a value $\epsilon = 1.8 \times 10^{-3}$ for a $\pi$ pulse applied on the $|1\rangle - |2\rangle$ transition; this is obtained by a best-fit of the $\theta = \pi$ data. For arbitrary $\theta$ it is natural to consider a linear interpolation $\epsilon[\theta] = 1.8 \times 10^{-3} \times \theta/\pi$.

## Data availability
Experimental and simulated data generated during this study are included in this published article (and its supplementary information files). The experimental data that support the findings of this study can also be found in the GitHub repository[65].

## Code availability
The codes for simulations that support the findings of this study can be found in the GitHub repository[65].

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

## Acknowledgements
We are grateful to Kirill Petrovnin, Aidar Sultanov, Andrey Lebedev, Sergey Danilin, and Miika Haataja for assistance with sample fabrication and measurements. This project has received funding from the European Union's Horizon 2020 research and innovation programme under grant agreement no. 862644 (FET-Open project QUARTET). We also acknowledge support from the Academy of Finland under the RADDESS programme (project 328193) and the Finnish Center of Excellence in Quantum Technology QTF (projects 312296, 336810), as well as from Business Finland QuTI (decision 41419/31/2020). This work used the experimental facilities of the Low Temperature Laboratory and Micronova of OtaNano research infrastructure.

## Author contributions
S.D. and G.S.P. conceived the idea and obtained the key results. S.D. performed the experiments and did a detailed analysis of the experimental data with inputs from J.J.M. S.D. and J.J.M. did the numerical simulations. G.S.P. supervised the project. All authors contributed to analytical calculations, discussed the results, and wrote the manuscript.

## Competing interests
The authors declare no competing interests.
