## [Peer Review File · Nature Communications]

REVIEWER COMMENTS

Reviewer #1 (Remarks to the Author):

I have carefully read through the manuscript "Coherent interaction-free detection of microwave pulses with a superconducting circuit".

My overall impression of the results and their importance is quite positive, and I am generally very favorable towards publication. The subject is timely and important, and the results are convincing and of a broad and general interest.

However, I find the presentation, flow and overall clarity of the manuscript seriously flawed, and would like to recommend a comprehensive restructuring prior to publication.

Also, there are some important points that need to be discussed further, in order to clarify the utility and scope of the results.

Missing points that need improvement :

1. The introduction is not sufficiently broad/motivating, and should also include explicitly: nested interferometer measurements, partial and weak measurement, trajectory (quantum jump) perspective.
2. The paper peters out without a strong message/conclusion at the end. What is missing is a quantitative comparison/graph of how this "detector" compares with state of the art in superconducting photon detection.
3. What is unclear, and should be clarified, is the bandwidth of this "detector". Related issues are the ultimate sensitivity/noise properties which should be more explicitly stated for this scheme (large N , depolarizing imperfections ϵ , qubit lifetimes T_1 and T_2).

Restructuring issues:

0. I generally like the $N=1$, $N=2$ and $N>2$ presentation, but the $N=2$ part doesn't really add much clarity and maybe can be moved to the supplementary information.
1. In general, the manuscript is very heavy with definitions and subsequent internal jargon, and this should be improved/minimized if possible. For example, the definition paragraph of the "confusion

matrix" needs to be replaced by some sort of table/graphic since it is quite opaque to follow the text/definitions in the current form.

2. Related, the figure captions should be much more self-contained, and avoid jargon that requires multiple reading of the body text in order to understand what is being plotted.

3. Fig. 4 and the Majorana representation (in the supplementary information) should be improved and included in the main paper, since they are important and present a fresh (esp. the Majorana rep.) perspective.

Reviewer #2 (Remarks to the Author):

The manuscript by Dogra et al., presents a protocol used to detect MW pulses without absorbing the MW pulse. This protocol relies on the coherent evolution of a qutrit to enhance the detection efficiency in comparison to a projective protocol. The authors then implemented this protocol in a superconducting transmon qutrit and compared their results with a master equation-based model. The figure of merit used by the authors is shown to improve the efficiency compared with the projective, "interaction-free" readout method.

I would first like to point out that the paper is very well written. The authors present the result in a clear and coherent style. I am very impressed with the quality of the work (both the main text and the supplementary) and would think this work worthy of a publication of some sort with only minor further editing. The use of superconducting transmon qutrit to demonstrate this effect is also a great choice, given how clean the system is and the ability of dispersive readout. The scientific conclusion is well justified by the data and simulation. The concept is novel and interesting. The technique is readily applicable to systems like NV centers in diamond where a qutrit is available and further to optomechanical systems.

I am hesitant to recommend this paper for publication in Nature Communications, not based on the quality of the paper, but more on its importance and relevance to the field. Below I list down my comments.

The "interaction-free" detection was discussed in the supplementary reference 1 cited by the authors (Kwiat, 1995) and reference to Dicke, 1981 therein. Although the use of coherent evolution is new in comparison to the Kwiat 1995 paper published in PRL, fundamentally the key idea is still that the measurements collapse the probed MW field to the same quantum state (without causing a transition). This thus allows "interaction-free" measurement. I find this idea elegant, but the results presented in the manuscript do not seem a significant enough addition to the Kwiat 1995 paper. Would it be more suitable for the authors to instead submit the paper to journals from the Physical Review family? These novel and anti-intuitive tests of quantum theories are well-suited to the audience of a PR family journal.

Secondly, the “interaction-free” condition as discussed in the supplementary equation 10 is that there is no transition between the Fock states of the quantized MW field. However, the interaction-free method is not the only method to realize this condition. In spin-based quantum sensing literature (for example Neumann, science 2010, Single-Shot Readout of a Single Nuclear Spin), a SzIz type Hamiltonian allows the spin state (m_I) of a neighboring nucleus to be determined without a spin flip on the nuclear spin. Could the authors comment on the various methods for non-destructive measurements and how the “interaction-free” technique compares with these methods?

I look forward to the authors’ response to these points and their convincing arguments for publication in Nature Communications, which cover a broad range of audience.

REVIEWER COMMENTS

Reviewer #1 (Remarks to the Author):

I have carefully read through the manuscript "Coherent interaction-free detection of microwave pulses with a superconducting circuit".

My overall impression of the results and their importance is quite positive, and I am generally very favorable towards publication. The subject is timely and important, and the results are convincing and of a broad and general interest.

However, I find the presentation, flow and overall clarity of the manuscript seriously flawed, and would like to recommend a comprehensive restructuring prior to publication.

Also, there are some important points that need to be discussed further, in order to clarify the utility and scope of the results.

Answer: We are grateful for the very positive comments on the importance of our results and indeed we agree that a restructuring of the paper and more clarifications and a unified terminology should improve the quality of the presentation. We have done our best in the present version. Some more specific issues are addressed below.

Missing points that need improvement :

1. The introduction is not sufficiently broad/motivating, and should also include explicitly: nested interferometer measurements, partial and weak measurement, trajectory (quantum jump) perspective.

Answer: We have expanded the introduction to include all of the above and more. Some connections for example with the POVM formalism are already made in the SI. We also believe that our results will stimulate plenty of subsequent theoretical work from different perspectives on our protocol.

2. The paper peters out without a strong message/conclusion at the end. What is missing is a quantitative comparison/graph of how this "detector" compares with state of the art in superconducting photon detection.

Answer: We have introduced in the present version a Discussion section, where we outline exactly these issues and we compare with other superconducting detection schemes to the extent that this is meaningful (since each scheme often has its own figure of merit and no other interaction-free experiment has been done with superconducting elements). For example, from the data provided in the papers we can estimate the dark count rate of 0.1 counts/microsec for the Inomata et al. experiment and 0.2 counts/microsec for the Besse et al. experiment; ours is 0.1 counts/microsec. It is very important to notice that for us there is a clear pathway of reducing the false positives (dark counts), which is not available in other schemes, where typically a reduction of the dark counts, if possible, is accompanied by a reduction in the true positives. In our case, the true positives rapidly reach a plateau where they still slightly increase with N , while we can in principle lower the false positives by reducing the decoherence and pulse errors in the 01 subspace for a large number of sequences N - which corresponds exactly to the roadmap of reducing single-qubit errors when many consecutive gates are applied (as typically diagnosed by randomized benchmarking).

We have also compared our protocol with the optical realizations – and we find out that our efficiencies are higher than what has been reported in the literature and much more robust to increasing values of N .

3. What is unclear, and should be clarified, is the bandwidth of this "detector". Related issues are the ultimate sensitivity/noise properties which should be more explicitly stated for this scheme (large N , depolarizing imperfections ϵ , qubit lifetimes T_1 and T_2).

Answer: The bandwidth of the detector is given by the inverse of the duration of the B-pulses employed in our experiments, which was 56 ns, so it is about 18 MHz, quite typical for superconducting-circuit based detectors. We also now discuss the other figures of merit pointed out above. The noise due to dark counts is more explicitly put in evidence. At the same time we do not want to overwhelm the reader with technical details that are specific to our experiments – therefore some are relegated to Methods (the section about sample specification, decoherence values for both transitions, etc), as we would like to emphasize the universality of our protocol.

Restructuring issues:

0. I generally like the $N=1$, $N=2$ and $N>2$ presentation, but the $N=2$ part doesn't really add much clarity and maybe can be moved to the supplementary information.

Answer: We have been pondering as well where the $N=2$ case belongs. In fact, in the first draft of this work, we had put it in the SI. But then, we thought there is some justification for having it in the main text, namely:

- The $N=2$ case is the first step where our coherent protocol becomes fundamentally different from the standard projective protocol. It is also a step where one can still do relatively simple analytical calculations and compare the two cases.
- The experimental data and the fitting show that we have good control of the system also when more than one sequence is applied.
- It motivates the analysis done later for $N>2$, whereby we cannot directly plot the results in an N -dimensional theta-parameter space

We have now made more clear these motivations in the text. However, we do not have a very strong opinion on this, and if the Referee requires us to move it in the SI we can do so easily.

1. In general, the manuscript is very heavy with definitions and subsequent internal jargon, and this should be improved/minimized if possible. For example, the definition paragraph of the "confusion matrix" needs to be replaced by some sort of table/graphic since it is quite opaque to follow the text/definitions in the current form.

We went through our terminology and we think we have improved on this aspect. For example, we were using "populations", which is now consistently replaced with "probabilities". We have defined now all the relevant notations at the end of the second section. About the confusion matrix, we provide now a better description in the text, and we have added a table in the SI.

2. Related, the figure captions should be much more self-contained, and avoid jargon that requires multiple reading of the body text in order to understand what is being plotted.

Answer: We have tried to streamline the captions and the explanations in the text. One issue is that we have a lot of data and there are several ways of representing them, since as $N > 1$ the figures of merit become multidimensional. So, on one hand we have to select the most important ones but at the same time we would like to show as much information as possible. We hope that in this version we struck the right balance.

3. Fig. 4 and the Majorana representation (in the supplementary information) should be improved and included in the main paper, since they are important and present a fresh (esp. the Majorana rep.) perspective.

Answer: We have added a better plot of the Majorana representation in the main paper. We have also added the definition and more information about the efficiency values in the main paper, as well as a comparison with the results obtained in optical setups. We prefer to just give some representative numbers and keep Fig. 4 (as well as Figs. 2 and 3) in the SI because we already have too many figures in the main text. But a more fundamental reason is that, seen from the perspective of detector theory as used in radars, etc., the relevant quantities are the positive and negative ratios. In other words, if we think about the bomb analogies, the data we end up with are those for the cases where the bomb did not explode. In other words, when used in practice as a non-absorbing pulse detector we would discard anyway the p_2 results and try to figure out if there was a pulse or not from the state 0 and 1 clicks.

Overall, we greatly appreciate the Referee's positive comments on the importance of our results, and we have restructured the paper with more clarifications added.

Reviewer #2 (Remarks to the Author):

The manuscript by Dogra et al., presents a protocol used to detect MW pulses without absorbing the MW pulse. This protocol relies on the coherent evolution of a qutrit to enhance the detection efficiency in comparison to a projective protocol. The authors then implemented this protocol in a superconducting transmon qutrit and compared their results with a master equation-based model. The figure of merit used by the authors is shown to improve the efficiency compared with the projective, "interaction-free" readout method.

I would first like to point out that the paper is very well written. The authors present the result in a clear and coherent style. I am very impressed with the quality of the work (both the main text and the supplementary) and would think this work worthy of a publication of some sort with only minor further editing. The use of superconducting transmon qutrit to demonstrate this effect is also a great choice, given how clean the system is and the ability of dispersive readout. The scientific conclusion is well justified by the data and simulation. The concept is novel and interesting. The technique is readily applicable to systems like NV centers in diamond where a qutrit is available and further to optomechanical systems.

I am hesitant to recommend this paper for publication in Nature Communications, not based on the quality of the paper, but more on its importance and relevance to the field. Below I list down my comments.

The "interaction-free" detection was discussed in the supplementary reference 1 cited by the authors (Kwiat, 1995) and reference to Dicke, 1981 therein. Although the use of coherent evolution is new in

comparison to the Kwiat 1995 paper published in PRL, fundamentally the key idea is still that the measurements collapse the probed MW field to the same quantum state (without causing a transition). This thus allows “interaction-free” measurement. I find this idea elegant, but the results presented in the manuscript do not seem a significant enough addition to the Kwiat 1995 paper. Would it be more suitable for the authors to instead submit the paper to journals from the Physical Review family? These novel and anti-intuitive tests of quantum theories are well-suited to the audience of a PR family journal.

Answer: We would like to thank the Referee for the kind words showing appreciation of our work. We hope that the quality of the paper has improved even more due to the present round of refereeing, where, as can be seen, we have added a number of clarifying discussions and results. Our protocol is not just a simple extension of the standard interaction-free projective protocol: it has fundamentally different features, which are also reflected in the final higher success probabilities. These high figures of merit are predicted theoretically and observed experimentally, and they are clearly superior to what has been achieved in optics, even by subsequent dedicated experiments using waveguides (Ma et. al. 2014). It demonstrates a type of quantum enhancement – by using the coherence of the qutrit system we get better results than by repeated collapse of the wavefunction by negative-events (no-absorption or no-bomb-explosion as in the original Mach-Zehnder gedankenexperiment). We believe that the protocol we have proposed, and its experimental validation are of interest for the general scientific community, and will attract the attention of both theorists and experimentalists. For example, one can further start to investigate questions such as: are there implications for quantum estimation – how precisely can one estimate θ based on this method, how does this precision scale with the number of measurements? Or, how does the method generalize to sensing of photons in multiple cavities? Many additional connections with other fields of research exist and are now mentioned in the introduction. Besides fundamental science, applications would be top of the list. For example, our group is interested in optimizing our system for axion search (detection of microwave photons emitted by the presumed axions, now mentioned at the end of the discussion section) – an application that demonstrates that our results are of broader interest. Other applications that can be immediately envisioned are in microwave fluorescence (e.g. photo-counting statistics fluorescence of rare-earth doped materials). We also emphasize that the protocol can be applied to any experimental platform where a three-level system and the associated detection is available.

Secondly, the “interaction-free” condition as discussed in the supplementary equation 10 is that there is no transition between the Fock states of the quantized MW field. However, the interaction-free method is not the only method to realize this condition. In spin-based quantum sensing literature (for example Neumann, science 2010, Single-Shot Readout of a Single Nuclear Spin), a SzIz type Hamiltonian allows the spin state (mI) of a neighboring nucleus to be determined without a spin flip on the nuclear spin. Could the authors comment on the various methods for non-destructive measurements and how the “interaction-free” technique compares with these methods?

Answer: We are now providing a more in-depth comparison and we have added the relevant literature including the spin-based sensing experiment above. In brief, our protocol is in a rather different class, known as “hypothesis testing” while the protocol implemented in the paper above is a von Neumann QND measurement. For example, a radar is a hypothesis testing system - which detects the existence or absence of a target but is not necessarily concerned with detailed imaging of that target – while in the case of von Neumann measurements one can use them to do state tomography.

In general, for any protocol quantum physics tells us that eventually we will obtain a state where the pointer is entangled with the state of the system. We can see this both for our protocol (see e.g. Eq.

13) and for the protocol elaborated in Neumann et. al., where the entanglement is achieved by a CNOT gate. This gate is constructed by flipping the state of the electron spin with an external microwave field depending on whether the nuclear spin is in $|n\rangle$. Thus, one injects energy into the system. In our protocol we do not have this conditional operation: rather, what happens is that the amplitude probabilities in the 0-1 subspace are changed, which modifies the result of the Ramsey interference. In our protocol, the requirement is to use only the interaction Hamiltonian Eq. (10) at resonance. This conserves the number of excitations, in other words what we can do is an iSWAP gate with the second transition, starting with the transmon in state $|1\rangle$, and resulting in a state of the type $|1\rangle|n\rangle + |n-1\rangle|2\rangle$. Now, the only way we can claim to detect the photons is by observing state $|2\rangle$; otherwise, if we observe state $|1\rangle$ it means that the box was not present. But then the observation results in an absorption of a photon from the box into the transmon. One can still try to go around this problem by constructing a CNOT gate (with the control on the photon and the target on the transmon's $\{|1\rangle, |2\rangle\}$ subspace) from the iSWAP, in order to recreate something as close as possible to the standard von Neumann measurement model (as used in the Science paper). Let us assume for simplicity that $n=0,1$. Then the state $|1\rangle|n=0\rangle$ (no photons or no cavity) will be mapped onto the same state, and the state $|1\rangle|n=1\rangle$ (cavity present with a photon in) will be mapped onto $|2\rangle|n=1\rangle$, so it looks like we have achieved detection without absorbing the photon. However, the CNOT can be realized from iSWAP only with the addition of single-qubit gates, in other words again by injecting and extracting energy from the system. These observations are now included in the new Discussion section in the main text as well as at the end of Supplementary Note 2, where it is worked out in detail.

I look forward to the authors' response to these points and their convincing arguments for publication in Nature Communications, which cover a broad range of audience.

Answer: We are hopeful that the present version brings out more clearly the importance and wide applicability of our results.

REVIEWERS' COMMENTS

Reviewer #1 (Remarks to the Author):

The authors have complied with all my requests and I find the paper well motivated and novel, with a reasonable clarity and flow. Jargon and figure caption issues have been addressed properly and I recommend publication.

Personally, I would still move the bulk of the $N=2$ case description to supplementary information, to improve the clarity and flow of the manuscript. However, I agree with the authors this is a style/presentation decision within their mandate, and I don't insist.

Reviewer #3 (Remarks to the Author):

The authors experimentally demonstrated a protocol using repeated coherent interrogations instead of the projective ones. In this protocol, a series of N Ramsey-like sequences are resonant with the transition between the ground state and the first excited state of a transmon qutrit, while certain microwave pulses called the B -pulses in the manuscript resonantly couple to the transition between the first and second excited states. Its goal is to detect microwave B -pulses without absorbing them. In my opinion, this protocol is novel and presents useful results, which can be extended to other three-level systems. It should undoubtedly be of interest to a wide audience in the community.

In short, referee 2 raised two important points. In the first point, it is mentioned that the use of coherent evolution is new in comparison with previous studies, but the key idea is fundamentally still that the measurements collapse the probed microwave pulses to the given state without causing a transition, namely, it is related to the interaction-free projective measurement. The authors explained in detail that their protocol is not just a simple extension of the standard interaction-free projective measurement, but presents fundamentally different features, including that reflected in the final higher success probabilities.

In the second point, referee 2 mentioned that the interaction-free method is not the only method to realize the “interaction-free” condition discussed in the supplementary material and listed some relevant papers. The authors explained that the protocol implemented in these papers is a von Neumann quantum non-demolition (QND) measurement, while their protocol involves “hypothesis testing” and belongs to a rather different class.

I agree with these explanations and the novelty of their protocol in comparison with previous ones assures its quality meeting the criteria of Nature Communications. Also, it can be seen that the manuscript is considerably revised and much improved by considering the comments and suggestions made by the two referees. Thus, I think it is ready to be publishable after a minor revision listed below.

The protocol in this manuscript can involve considerable quantum operations. I think its experimental success is attributed to the high coherence of the transmon qubit used when considering the impact of the decoherence in the system. In fact, the transmon is a capacitively shunted Josephson junction, in which the large capacitor shunted to the Josephson junction plays the essential role of reducing the sensitivity of the device to the charge noise [PRB 75, 140515 (2007); PRA 76, 042319 (2007); PRL 111, 080502 (2013)]. This yields the transmon to have sufficiently high coherence for implementing the protocol. To have the transmon tunable, the junction can be replaced by a SQUID, as done in the manuscript. However, the manuscript does not have the key description of the transmon related to the function of the large shunt capacitance. There is no need to ignore the important credit of these papers in implementing a high-coherence superconducting qubit.

REVIEWERS' COMMENTS

Reviewer #1 (Remarks to the Author):

The authors have complied with all my requests and I find the paper well motivated and novel, with a reasonable clarity and flow. Jargon and figure caption issues have been addressed properly and I recommend publication.

Personally, I would still move the bulk of the $N=2$ case description to supplementary information, to improve the clarity and flow of the manuscript. However, I agree with the authors this is a style/presentation decision within their mandate, and I don't insist.

We wish to thank the Referee for the positive feedback and are delighted that publication is recommended.

We appreciate the Referee's suggestion of moving the $N = 2$ case to the supplementary information. We have also weighted the pros and cons, and in the end we thought it offers a better flow of the narrative, for the following reasons (mentioned also in the text):

- The $N=2$ case is the first step where our coherent protocol becomes fundamentally different from the standard projective protocol. It is also a step where one can still do relatively simple analytical calculations and compare the two cases.
- The experimental data and the fitting show that we have good control of the system also when more than one sequence is applied.
- It motivates the analysis done later for $N>2$, whereby we cannot directly plot the results in an N -dimensional theta-parameter space.

Reviewer #3 (Remarks to the Author):

The authors experimentally demonstrated a protocol using repeated coherent interrogations instead of the projective ones. In this protocol, a series of N Ramsey-like sequences are resonant with the transition between the ground state and the first excited state of a transmon qutrit, while certain microwave pulses called the B -pulses in the manuscript resonantly couple to the transition between the first and second excited states. Its goal is to detect microwave B -pulses without absorbing them. In my opinion, this protocol is novel and presents useful results, which can be extended to other three-level systems. It should undoubtedly be of interest to a wide audience in the community.

In short, referee 2 raised two important points. In the first point, it is mentioned that the use of coherent evolution is new in comparison with previous studies, but the key idea is fundamentally still that the measurements collapse the probed microwave pulses to the given state without causing a transition, namely, it is related to the interaction-free projective measurement. The authors explained in detail that their protocol is not just a simple extension of the standard interaction-free projective measurement, but presents fundamentally different features, including that reflected in the final higher success probabilities.

In the second point, referee 2 mentioned that the interaction-free method is not the only method to realize the “interaction-free” condition discussed in the supplementary material and listed some relevant papers. The authors explained that the protocol implemented in these papers is a von Neumann quantum non-demolition (QND) measurement, while their protocol involves “hypothesis testing” and belongs to a rather different class.

I agree with these explanations and the novelty of their protocol in comparison with previous ones assures its quality meeting the criteria of Nature Communications. Also, it can be seen that the manuscript is considerably revised and much improved by considering the comments and suggestions made by the two referees. Thus, I think it is ready to be publishable after a minor revision listed below.

We thank the Referee for thoroughly reviewing our manuscript and are pleased with the favorable response towards publication.

The protocol in this manuscript can involve considerable quantum operations. I think its experimental success is attributed to the high coherence of the transmon qubit used when considering the impact of the decoherence in the system. In fact, the transmon is a capacitively shunted Josephson junction, in which the large capacitor shunted to the Josephson junction plays the essential role of reducing the sensitivity of the device to the charge noise [PRB 75, 140515 (2007); PRA 76, 042319 (2007); PRL 111, 080502 (2013)]. This yields the transmon to have sufficiently high coherence for implementing the protocol. To have the transmon tunable, the junction can be replaced by a SQUID, as done in the manuscript. However, the manuscript does not have the key description of the transmon related to the function of the large shunt capacitance. There is no need to ignore the important credit of these papers in implementing a high-coherence superconducting qubit.

We agree that it is the capacitively shunted SQUID which allows for the high coherence required by our protocol, and that it effectively reduces the transmon’s sensitivity to charge noise. We have now included a small paragraph making this important point clear, and have cited the three papers recommended by the Referee.